# ACE2 and TMPRSS2 Immunolocalization and COVID-19-Related Thyroid Disorder

**DOI:** 10.3390/biology11050697

**Published:** 2022-04-30

**Authors:** Gi-Cheol Park, Hyoun-Wook Lee, Ji-Min Kim, Ji-Min Han, Hye-In Kim, Sung-Chan Shin, Yong-il Cheon, Eui-Suk Sung, Minhyung Lee, Jin-Choon Lee, Dong-Min Shin, Byung-Joo Lee

**Affiliations:** 1Department of Otolaryngology—Head and Neck Surgery, Samsung Changwon Hospital, Sungkyunkwan University School of Medicine, Changwon 51353, Korea; uuhent@gmail.com (G.-C.P.); bluesky61u@naver.com (D.-M.S.); 2Department of Pathology, Samsung Changwon Hospital, Sungkyunkwan University School of Medicine, Changwon 51353, Korea; sudowo@naver.com; 3Department of Otorhinolaryngology—Head and Neck Surgery, Pusan National University Hospital, Biomedical Research Institute, College of Medicine, Pusan National University, Busan 49241, Korea; ny5thav@hanmail.net (J.-M.K.); cha-nwi@hanmail.net (S.-C.S.); skydragonone@naver.com (Y.-i.C.); 4Department of Medicine, Division of Endocrinology and Metabolism, Samsung Changwon Hospital, Sungkyunkwan University School of Medicine, Changwon 51353, Korea; jimin0306@gmail.com (J.-M.H.); hyein11.kim@gmail.com (H.-I.K.); 5Department of Otorhinolaryngology—Head and Neck Surgery, Pusan National University Yangsan Hospital, Biomedical Research Institute, College of Medicine, Pusan National University, Yangsan 50612, Korea; sunges77@gmail.com (E.-S.S.); weichwein@naver.com (M.L.); ljc0209@hanmail.net (J.-C.L.)

**Keywords:** ACE2, TMPRSS2, COVID-19, SARS-CoV-2, thyroid

## Abstract

**Simple Summary:**

We analyzed the underlying mechanism of thyroidal manifestations in patients with COVID-19 using ACE2 and TMPRSS2 immunostaining. Prior to our work, few studies had investigated the mRNA expression of ACE2 in the thyroid gland. However, there has been no tissue-level study based on histological staining of ACE2 and TMPRSS2. Here, we found for the first time that ACE2 and TMPRSS2 proteins were not expressed in thyroid follicular cells and that ACE2 was exclusively expressed in thyroidal pericytes by immunolocalization of ACE2 and TMPRSS2 in the human thyroid tissue. The results of the present study demonstrate that SARS-CoV-2 does not directly invade thyroid follicular cells, but microcirculatory damage caused by pericyte infection may affect surrounding follicles in the thyroid gland.

**Abstract:**

Thyroid dysfunction has been reported to be an extrapulmonary symptom of COVID-19. It is important to identify the tissue subset that expresses angiotensin-converting enzyme 2 (ACE2) and transmembrane protease serine 2 (TMPRSS2), which are essential for host infection with severe acute respiratory syndrome coronavirus 2 (SARS-CoV-2), in order to understand the viral pathogenesis of COVID-19-related thyroid dysfunction. We investigated the expression and distribution of ACE2- and TMPRSS2-expressing cells in the thyroid gland. RT-PCR and Western blotting were performed on human thyroid follicular cells (Nthy-ori3-1) and rat thyroid tissues to detect the expression levels of ACE and TMPRSS2 mRNA and proteins. We also analyzed the expression patterns of ACE2 and TMPRSS2 in 9 Sprague-Dawley rats and 15 human thyroid tissues, including 5 normal, 5 with Hashimoto’s thyroiditis, and 5 with Graves’ disease, by immunohistochemistry (IHC) and immunofluorescence. Both ACE2 and TMPRSS2 mRNAs and proteins were detected in the thyroid tissue. However, ACE2 and TMPRSS2 proteins were not expressed in thyroid follicular cells. In IHC, ACE2 and TMPRSS2 were not stained in the follicular cells. No cells co-expressed ACE2 and TMPRSS2. ACE2 was expressed in pericytes between follicles, and TMPRSS2 was mainly stained in the colloid inside the follicle. There was no difference in expression between the normal thyroid, Hashimoto’s thyroiditis, and Graves’ disease. SARS-CoV-2 does not directly invade the thyroid follicular cells. Whether SARS-CoV-2 infection of pericytes can affect COVID-19-related thyroid dysfunction warrants further study.

## 1. Introduction

The COVID-19 pandemic, which has swept the world since it was first reported in December 2019, is still ongoing. COVID-19 mainly causes fever, dry cough, and fatigue, but can range from asymptomaticity to pneumonia and, in severe cases, fatal respiratory damage [1,2]. According to several recent reports, COVID-19 affects not only the respiratory system but also various organs and can cause cardiovascular symptoms such as chest stress and cardiac injury; digestive symptoms such as diarrhea, nausea, and vomiting; and neurologic symptoms such as headache or confusion [3,4,5].

The severe acute respiratory syndrome coronavirus 2 (SARS-CoV-2) causing COVID-19 penetrates the host cell by using the spike (S) protein protruding from the surface of the envelope [6]. The S protein consists of an S1 subunit that has a receptor-binding motif and interacts directly with the host cell receptor, as well as an S2 subunit that fuses with the cell of the host mucosa. The S1 subunit recognizes a membrane protein, the angiotensin-converting enzyme 2 (ACE2). Once bound to ACE2, another membrane protein, transmembrane protease serine 2 (TMPRSS2), cuts the S protein at a specific location between the S1 and S2 subunits. The cleaved S1 subunit moves away with the ACE2, and the remaining S2 subunit enables the fusion of the host cell membrane and the viral envelope. Although several other membrane proteins are involved in this process, ACE2 and TMPRSS2 are well known as the two most important proteins for SARS-CoV-2 invasion [7,8,9]. So, tissues expressing both ACE2 and TMPRSS2 can be the potential targets of SARS-CoV-2. ACE2 and TMPRSS2 are reported to be expressed not only in the respiratory tract, but also in the gastrointestinal mucosa, myocardium, vessels, and nerves [8,9,10]. These findings suggest that SARS-CoV-2 can directly invade other tissues as well as the respiratory mucosa.

During the COVID-19 pandemic, COVID-19-related thyroid dysfunctions have been reported, such as non-thyroidal illness, hypothyroidism, painless thyroiditis, subacute thyroiditis, and Graves’ disease [11]. However, it is still unknown whether COVID-19-related thyroid disorder is caused by direct SARS-CoV-2 invasion of the thyroid gland. Studies using RNA expression databases or quantitative reverse transcription (RT)–polymerase chain reaction (PCR) in in vitro ex vivo studies on thyroid have reported that ACE2 and TMPRSS2 are expressed in the thyroid, suggesting that the thyroid could be a potential target for SARS-CoV-2 [10,12,13]. However, there has been no tissue-level study through ACE2 and TMPRSS2 histologic staining. RNA expression databases cannot localize ACE2 and TMPRSS2 in the thyroid.

The thyroid gland consists not only of follicular cells, but also parafollicular C cells, colloid, vessels, and nerves. To confirm the pathogenesis of thyroidal manifestations of COVID-19, it is essential to localize ACE2 and TMPRESS2 expression in the thyroid. Therefore, we investigated the expression and distribution of ACE2 and TMPRSS2 in the thyroid gland to identify the histological mechanism underlying thyroid disorder-related COVID-19.

## 2. Materials and Methods

### 2.1. Rat Thyroid Tissue

Nine female Sprague-Dawley rats (Samtako, Osan, Korea) were used in this study. Animal care and research protocols were based on the principles and guidelines of the Guide for the Care and Use of Laboratory Animals. Rats were euthanized after a week of acclimatization, as described in our previous study [14]. Thyroid tissue samples were isolated from each rat and fixed one week in 4% formalin. Paraffin embedding performed by automatic tissue processor for paraffin embedding (TP1020, Leica, Wetzlar, Germany) and dispensing (EG1150H, Leica). Cross-sections (8 μm thick) were placed on glass slides and prepared for hematoxylin–eosin staining. For staining analyses, the slides were de-paraffinized with xylene and then hydrated through a series of washes in ethanol and finally pure water. We selected the central part of the tissue for representative images using undertaken at 200× pictures by light microscope (Leica, Wetzlar, Germany, Leica DM4000/600M, versatile upright microscope for materials analysis).

### 2.2. Human Thyroid Cell Line

Commercially available human thyroid follicular cell line Nthy-ori 3-1 (https://web.expasy.org/cellosaurus/CVCL_2659.txt; accessed on 10 December 2021) was cultured in Roswell Park Memorial Institute medium 1640 with 10% fetal bovine serum, 2 mM glutamine, and antibiotics (100 U/mL penicillin–streptomycin (Gibco)).

### 2.3. Patients and Tissue Preparation

Fifteen female thyroid tissues, including five normal, five Hashimoto’s thyroiditis, and five Graves’ disease tissues, were obtained from surgical specimens following thyroidectomy at our institution from 2015 to October 2021. Normal thyroid tissue was obtained from disease-free tissue in the contralateral lobe of a patient with unilateral papillary thyroid carcinoma (PTC), and Hashimoto’s thyroiditis tissue was obtained from tumor-free tissue of PTC patients with coexisting Hashimoto’s thyroiditis after surgery. Pathologically proven Hashimoto’s thyroiditis was defined as the presence of diffuse lymphoplasmacytic infiltrate, formation of lymphoid follicles with germinal centers, oxyphilic follicular cells, and atrophic changes in the area of normal thyroid tissue. Whole sections of formalin-fixed, paraffin-embedded patient tissue blocks were used for immunohistochemistry (IHC). The immunohistochemical results were interpreted by a board-certified pathologist (H.W.L.) who has more than 15 years of experience with thyroid head and neck tumors.

### 2.4. Quantitative Reverse Transcription–Polymerase Chain Reaction

The tissue and cells were rinsed with cold PBS, and RNA was extracted using the TRIzol system (Life Technologies, Rockville, MD, USA). cDNA synthesis was performed using a reverse transcription kit (Applied Biosystems, Foster City, CA, USA), in accordance with the manufacturer’s protocol. RT-PCR was performed to determine the mRNA expression levels of ACE2, TMPRSS2, β-actin, and GAPDH, which were used as housekeeping controls. Primer sequences are shown in Appendix A. Real-time quantification was based on the LightCycler assay using a fluorogenic SYBR Green I PCR mixture on a LightCycler instrument (Roche, Mannheim, Germany). LightCycler version 3.3 software was used to analyze the PCR kinetics and quantitative data. All experiments were conducted three times, and negative and positive controls were included.

### 2.5. Western Blot

The tissue and cells were rinsed twice with cold PBS and lysed in radioimmunoprecipitation assay (RIPA) buffer (Sigma-Aldrich). Proteins were loaded onto a 10% sodium dodecyl sulfate-polyacrylamide (SDS) gel and electrotransferred onto polyvinylidene fluoride (PVDF) membranes (Millipore, MA, USA). The membrane was immediately placed in a blocking buffer (2% bovine serum albumin (BSA)) for 1 h. The membrane was incubated with anti-ACE2 (1:1000, Abcam, Cambridge, UK) and TMPRSS2 (Santa Cruz, 1:2000) at 4 °C overnight, using an anti-β-actin antibody (Santa Cruz, 1:2000) as a housekeeping control. After three 10 min washes, the membranes were incubated with m-IgGκ BP-HRP (1:10,000 dilution) for 1 h at room temperature. Antibody labeling was detected using West-Zol Plus and chemiluminescence FluorChem™ SP (Alpha Innotech Corporation, San Leandro, CA, USA).

### 2.6. Immunohistochemistry

Paraffin sections were deparaffinized with xylene and washed with phosphate-buffered saline (PBS). Sections were blocked with 2% BSA containing 0.3% Triton X-100 in PBS. They were then incubated for 24 h at 4 °C with the following primary antibodies: anti-ACE2 (1 mg/mL) (ab272690, Abcam) and anti-TMPRSS2 (1 μg/μL) (Bioss Antibodies Inc. sc-515727, Woburn, MA, USA). The next day, the primary antibody was removed by rinsing, and sections were incubated with secondary antibodies for 1 h at 21 °C. Goat anti-rabbit secondary antibodies (1:1000) (ENZO Biochem, NY, USA) were used for double staining with DAB (3, 3-diaminobenzidine) staining. The oral mucosa was stained as a positive control for ACE2 and TMPRSS2 (Appendix A). Incubation with PBS supplemented with 1% BSA instead of the primary antibody served as a negative control.

### 2.7. Immunofluorescence

For immunofluorescence microscopy, frozen blocks were thawed in PBS to enhance penetration and were blocked with 1% BSA for 30 min to inactivate secondary antibody binding sites. The cryo-sections were incubated overnight in the anti-ACE2 (1 mg/mL) (Abcam) and anti-NG2 (nerve/glial antigen 2) (1 mg/mL) (Invitrogen, Waltham, MA, USA) at 4 °C. After three washes with PBS (3 min, each), the slides were incubated with the secondary antibodies for 30 min. The next day, slides were incubated with FITC- or TRITC-conjugated secondary antibodies for 30 min at 21 °C. Images were acquired by confocal microscope (LAS-X).

All procedures using thyroid follicular cells and human and animal thyroid tissues were reviewed, approved, and conducted in accordance with the guidelines of our Institutional Animal Care and Use Committee and Institutional Review Board.

## 3. Results

### 3.1. ACE2 and TMPRSS2 Expression in the Thyroid Follicle

RT-qPCR was conducted to detect mRNA expression levels of ACE2 and TMPRSS2 in the whole rat thyroid. Both ACE2 and TMPRSS2 mRNAs were detected in all rat thyroid tissues. The transcript expression levels of ACE2 and TMPRSS2 compared with those of β-actin and GAPDH are shown in Figure 1A. In Western blot analysis (File S1), both ACE2 and TMPRSS2 proteins were expressed, but their expression was weaker than that in the reference kidney (Figure 1B). To confirm ACE2 and TMPRSS2 expression in follicular cells, the human follicular cell line Nthy-ori 3-1 was also evaluated. In PCR, ACE2 and TMPRSS2 mRNAs were expressed in Nthy-ori 3-1, but the Western blot revealed no protein expression of ACE2 and TMPRSS2 (Figure 1). The same results were obtained in all three experiments. These results suggest that ACE2 and TMPRSS2 proteins are expressed in tissue subsets other than follicular cells in the thyroid gland.

### 3.2. Immunolocalization of ACE2 and TMPRSS2 in the Thyroid

To identify the localization of ACE2 and TMPRSS2 in the thyroid gland, IHC was performed. ACE2 and TMPRSS2 were sporadically expressed in different spots (Figure 2A). We observed that ACE2 was not stained in the follicular cells themselves, but in microvessels or lymphatics between the follicles at a higher magnification. TMPRSS2 was mainly stained in the colloid inside the follicle, but not in the follicular cells or microvessels (Figure 2B). TMPRSS2 staining was observed in relatively large vessels (arterioles/venules) located outside the thyroid gland (Figure 3). We confirmed that ACE2 is present in microvessels by observing that the expression of ACE2 was consistent with that of NG2, a pericyte marker, by immunofluorescence staining (Figure 4).

ACE2 and TMPRSS2 expression in the human thyroid was not significantly different from that in the rat thyroid. Graves’ disease has a larger colloid and more abundant microvessels around the follicle than those found in normal or Hashimoto’s thyroiditis-affected thyroids. ACE2 was stained only in microvessels in the normal thyroid, Hashimoto’s thyroiditis, and Graves’ disease. Expression seemed more abundant in Graves’ disease than in the normal or Hashimoto’s thyroiditis-affected thyroids because of the increased number of microvessels around the follicle. TMPRSS2 was stained in the colloid of the normal thyroid, Graves’ disease, and Hashimoto’s thyroiditis, and was more strongly expressed than in rats (Figure 5).

## 4. Discussion

In present study, we not only identified the expression of ACE2 and TMPRSS2 in thyroid follicular cells, but also investigated the histologic localization of ACE2 and TMPRSS2 proteins in the human thyroid gland. In thyroid follicular cells, the mRNA expression of both ACE2 and TMPRSS2 was identified, but no protein expression was found. In IHC performed on the whole thyroid tissue, neither ACE2 nor TMPRSS2 was expressed in thyroid follicle cells. ACE2 was stained only in the microvessels around the follicles, while TMPRSS2 was expressed in the colloid and vessels outside the thyroid gland. These results were similar in normal thyroids, Hashimoto thyroiditis, and Graves’ disease.

Few studies have investigated ACE2 expression in the thyroid gland [10,12,13]. However, all studies that have been published thus far have only investigated the mRNA expression of ACE2 with in silico studies using datasets from The Cancer Genome Atlas database or by culturing thyroid follicular cells. However, they did not investigate the expression of ACE2 at the protein level. The abundance of mRNA is generally known to have a positive correlation with that of protein, but the strength of the correlation varies widely depending on the subject and gene [15]. Discrepancy between mRNA and protein expression have been reported [15,16]. Therefore, the expression level of mRNA is only informative but not predictive for that of the protein. In the study about human diseases, it is essential to compare these parameters together, because proteins play the final role in cells. In present study, we identified the ACE2 and TMPRSS2 expression at the gene and protein levels in thyroid follicular cells and thyroid tissue using PCR and Western blotting, respectively. In particular, we revealed the immunolocalization of ACE2 and TMPRSS2 in the human thyroid gland.

ACE2 is well known to be expressed in the endothelium of blood vessels [17]. In our previous study investigating ACE2 expression in the salivary gland and trigerminal nerve, ACE2 was strongly expressed in the capillary endothelium [18]. The abundance of ACE2 in blood vessels may be associated with SARS-CoV-2-induced microcirculation damage causing vascular wall injury, blood clots, and endothelitis [19]. Recent studies have established that pericytes and perivascular cells in small vessels are major sites for ACE2 expression in several organs, including the heart, brain, and lungs [20,21,22]. Pericytes are vascular mural cells that enwrap the endothelial cells of capillaries. These cells interact with endothelial cells to regulate capillary blood flow and maintain microvascular integrity and immune function [23]. Damage to pericytes by SARS-CoV-2 may result in perivascular inflammation and occlusion of capillary blood flow, which may eventually cause tissue ischemia. Therefore, infected pericyte by SARS-CoV-2 can lead to impairment to the blood–brain barrier, resulting in neurological symptoms [20,24], and cardiac pericyte infection can cause acute cardiac injury [21,25,26]. Since ACE-expressing pericytes are distributed throughout the body along the capillaries, SARS-CoV-2-infected pericytes can theoretically affect not only the brain and heart but also all organs where capillaries are present [20,21,22,23]. In this study, we confirmed that ACE2 is strongly expressed in pericytes. Microcirculatory damage caused by pericyte infection may also affect surrounding follicles in the thyroid gland.

Several mechanisms have been suggested to underlie COVID-19-related thyroid disorder. First, it is possible that the thyroid disorder is caused by the alteration of the hypothalamic–thyroid axis due to the viral infection of the hypothalamic–pituitary system since the SARS genomic sequences were found in neurons of the hypothalamus [27]. Another possible mechanism is a secondary effect through systemic immune responses caused by SARS-CoV-2. SARS-CoV-2 infection can induce cytokine storm in which proinflammatory cytokines such as such as IL-6, IL-1β, and TNF-α are elevated. Specifically, a hyper-activation of the Th1/Th17 response has been reported in patients with autoimmune and drug-induced thyroiditis, and an increase in IL-6 was described in the progress of destructive thyroiditis [28]. Moreover, treatment of SASR-CoV-2 infection can lead to thyroid hormonal changes in the pituitary–thyroid axis feedback loop. A decrease in thyroid-stimulating hormone levels in patients was observed, which may be caused by chronic hypoxemia and steroid treatment [11]. Finally, potential direct SARS-CoV-2 infection of the thyroid follicles has also been suggested based on ACE2 mRNA expression in follicular cells [12]. However, our study revealed that SARS-CoV-2 does not directly invade the thyroid follicle by confirming that ACE2 and TMPRSS2 are not stained in follicular cells.

This study had some potential limitations. We indirectly demonstrated the possibility of SARS-CoV-2 infection through ACE2 and TMPRSS2 expression. It has been accepted that ACE2 and TMPRSS2 co-expression is an essential condition for SARS-CoV-2 infection, and studies have also raised the possibility of SARS-CoV-2 infection based on ACE2 and TMPRSS2 expression. However, recently, along with occurrence of SARS-CoV-2 variants bearing a mutation of the S protein, another invasion mechanism through alternative membrane receptor other than ACE-2 receptor has been suggested [29]. Although ACE2 and TMPRSS2 are not expressed in thyroid follicle cells, the possibility of ACE2- and TMPRSS2-independent infection cannot be excluded. Recently, it has been reported that the SARS-CoV-2 nucleocapsid antigen was found in the thyroid follicular cells of some patients with COVID-19 [30]. We also know that this study alone is insufficient to prove the relationship between ACE2 expression in the pericyte and thyroid damage. To confirm that ACE2-expressing pericytes induce COVID-19-related thyroid disorders, it is necessary to simultaneously identify perivascular infection with SARS-CoV-2 and inflammation in the perivascular area of patients with COVID-19 presenting with thyroid dysfunction.

However, we found for the first time that ACE2 and TMPRSS2 proteins were not expressed in thyroid follicular cells and that ACE2 was exclusively expressed in thyroidal pericytes by immunolocalization of ACE2 and TMPRSS2 in the thyroid tissue. The effect of thyroidal pericyte infection with SARS-CoV-2 on the thyroid gland requires further research.

## 5. Conclusions

SARS-CoV-2 does not directly invade thyroid follicular cells. Whether SARS-CoV-2 infection of thyroidal pericytes is responsible for COVID-19-related thyroid dysfunction warrants further study.

## Figures and Tables

**Figure 1 biology-11-00697-f001:**
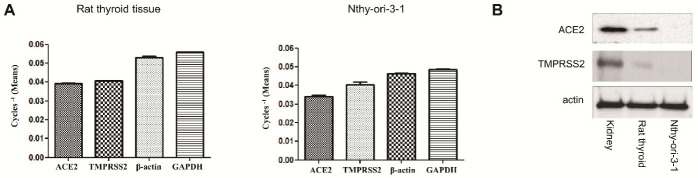
Expression of ACE2 and TMPRSS2 in the thyroid. (**A**) In RT-PCR, *ACE2* and *TMPRSS2* genes were expressed in both rat tissue and human follicular cells (Nthy-ori-3-1). The expression of ACE2, TMPRSS2, β-actin, and GAPDH are shown as means of cycles^−1^. (**B**) The Western blot revealed that protein expression of ACE2 and TMPRSS2 was detected in rat thyroid tissue, but not in follicular cells. *n* = 3. Columns and error bars represent the mean ± standard deviation. RT-PCR, quantitative reverse transcription–polymerase chain reaction; ACE2, angiotensin-converting enzyme 2; TMPRSS2, transmembrane protease serine; GAPDH, glyceraldehyde 3-phosphate dehydrogenase.

**Figure 2 biology-11-00697-f002:**
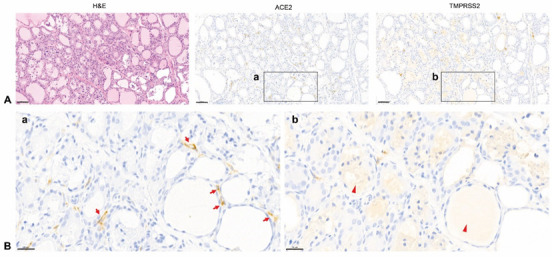
Immunolocalization of ACE2 and TMPRSS2 in the rat thyroid. (**A**) H&E staining (top left panel), ACE2 staining (top middle panel, brown), and TMPRSS2 staining (top right panel, brown). ACE2 and TMPRSS2 expression was observed sporadically in the whole thyroid tissue, but was not co-expressed. Scale bar = 50 μm. (**B**) At a higher magnification, ACE2 was not stained in follicular cells but in structures that appeared to be microvessels or lymphatics (arrow) between each follicle (**a**). TMPRSS2 was mainly expressed in the colloid (arrowhead) inside the follicle, but not in the follicular cells nor microvessles (**b**). Scale bar = 20 μm. H&E, hematoxylin and eosin; ACE2, angiotensin-converting enzyme 2; TMPRSS2, transmembrane protease serine.

**Figure 3 biology-11-00697-f003:**
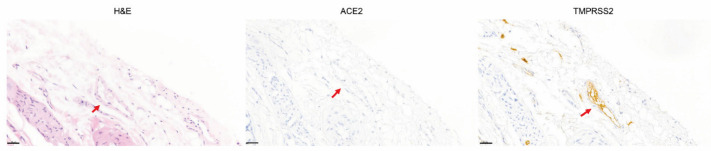
Expression of TMPRSS2 in the rat thyroid. Only TMPRSS2, not ACE2, was stained in relatively large vessels (arteriole/venule) (arrow) outside the rat thyroid gland. Scale bar = 20 μm. (**H&E**), hematoxylin and eosin; (**ACE2**), angiotensin-converting enzyme 2; (**TMPRSS2**), transmembrane protease serine.

**Figure 4 biology-11-00697-f004:**
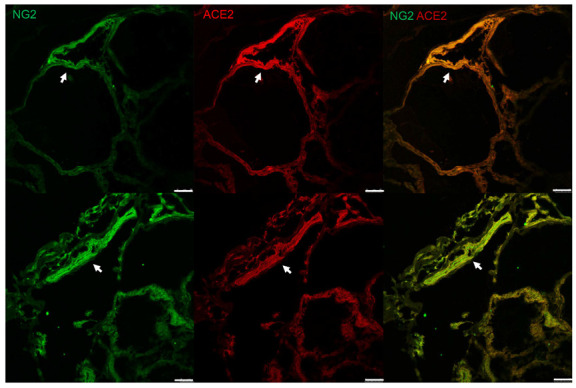
Expression of ACE2 in pericytes. Immunofluorescence staining showed that ACE2 was expressed in the interfollicular microvessels (arrow), and localization of ACE2 (middle panel, red) coincided with the expression of the pericyte marker NG2 (left panel, green). Scale bar = 20 μm. ACE2, angiotensin-converting enzyme 2; NG2, nerve/glial antigen 2.

**Figure 5 biology-11-00697-f005:**
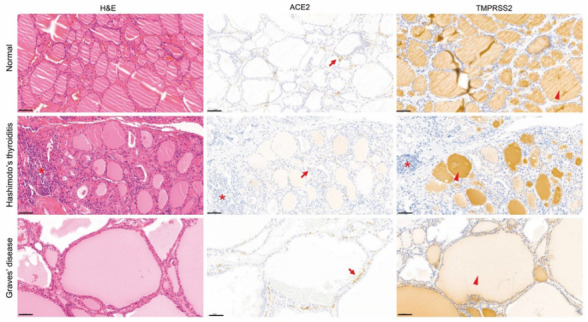
Immunolocalization of ACE2 and TMPRSS2 in the human thyroid. H&E staining ((**left**) panels), ACE2 staining ((**middle**) panels, brown), and TMPRSS2 staining ((**right**) panels, brown). In all three groups, including normal ((**top**) panels), Hashimoto’s thyroiditis ((**middle**) panels), and Graves’ disease ((**bottom**) panels), ACE2 was expressed only in the microvessels (arrow) between the follicles, but not in the follicular cells. TMPRSS2 was stained only in the colloid (arrowhead), although there was a difference in intensity in all three groups. Lymphocytes (asterisk) were observed in Hashimoto’s thyroiditis. The colloid was significantly greater in Graves’ disease than in the other two groups. Scale bar = 50 μm. H&E, hematoxylin and eosin; ACE2, angiotensin-converting enzyme 2; TMPRSS2, transmembrane protease serine.

## Data Availability

Not applicable.

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
