# Peer review of "ACE2 and TMPRSS2 Immunolocalization and COVID-19-Related Thyroid Disorder"

_biology, 2022, doi:10.3390/biology11050697_

Round 1
Reviewer 1 Report
This is an interesting study and a well-written paper. The authors bring a novelty by discovering the exact localization of ACE2 and TMPRSS2 proteins in thyroid tissue. The authors did a good job in the discussion part of the paper trying to explain the relationship and mechanisms of ACE2 expression and the harmful effects of COVID-19 on various tissues and organs such as the brain and heart, pointing out the hypothesis of direct and indirect possible effects of COVID-19 on thyroid tissue.
- What is the main question addressed by the research?
Following recent reports of thyroid deterioration during COVID-19 infection, the authors sought to demonstrate the mechanisms by which COVID-19 adversely affects the gland by targeting the expression and location of ACE2 and TMPRSS2 in the gland tissue, the components required for virus infection.
- Do you consider the topic original or relevant in the field, and if
so, why?
I think that the topic is original and relevant because it brings novelty. In this paper, it was pointed out that COVID-19 indirectly attacks the thyroid gland, considering that follicular cells do not produce ACE 2 and TMPRSS2 proteins, but cells of the cardiovascular system, pericyte.
3. What does it add to the subject area compared with other published
material?
This paper brings novelty of the exact location of ACE 2 and TMPRSS2 proteins in the thyroid gland, information never previously published as far as I know. In addition, the authors did a pretty good job in the discussion part of the manuscript, deeply arguing their funding, and providing possible explanations of mechanisms and hypotheses related to the negative impact of the COVID-19 on the thyroid gland.
- What specific improvements could the authors consider regarding the
methodology?
I do not have anything to add. Also, the authors pointed out by themselves the limitations of the study that future experiments and studies should respond to.
- Are the conclusions consistent with the evidence and arguments
presented and do they address the main question posed?
Yes, they are.
- Are the references appropriate?
Yes, they are.
- Please include any additional comments on the tables and figures.
- I do not have additional comments.
Author Response
Response to the Reviewer #1
Comments to the Author
I do not have anything to add. Also, the authors pointed out by themselves the limitations of the study that future experiments and studies should respond to.
Answer: We appreciate your kind comments.
Reviewer 2 Report
1. What is the main question addressed by the research? The main question is what is the expression pattern of ACE2 and TMPRSS in the thyroid in both the gene and protein level and could it explain thyroid distinction associated with COVID-19. 2. Do you consider the topic original or relevant in the field, and if so, why? The study is original, addressing ACE2 and TMPRSS expression in the protein level and localization to specific cells and morphologies. 3. What does it add to the subject area compared with other published material? Potential mechanisms are well discussed, addressing clinical aspects 4. What specific improvements could the authors consider regarding the methodology? The manuscript is suitable for publication, I wouldn’t recommend any changes 5. Are the conclusions consistent with the evidence and arguments presented and do they address the main question posed? Yes 6. Are the references appropriate? Yes 7. Please include any additional comments on the tables and figures. No additional comment, good luck
Author Response
Response to the Reviewer #2
Comments to the Author
The manuscript entitled “ACE2 and TMPRSS2 immunolocalization and COVID-19-re-lated thyroid disorder” by Gi Cheol Park et al is aimed at characterization ACE2 and TMPRSS2 expression in the thyroid gland, in order to link it to disorders associated with COVID-19. Gene expression analyses in the whole thyroid are followed by IHC and immunofluorescence, and demonstrate that both proteins are expressed in the thyroid, but not in the follicular cells. ACE2 is expressed between the follicles, and the two proteins are probably not co-expressed. Localization of ACE2 and TMPRSS2 expression by histological methods is novel and essential for understanding its role in SARS-CoV-2 pathogenicity. The clinical relevance is well discussed. The is one minor correction, “TMPRSS” in the second line of the Simple Summary should be changed to “TMPRSS2” Altogether I would like to recommend publishing this manuscript in Biology. Good luck
Answer: We appreciate your kind comments. As suggested, we have corrected “TMPRSS” in the second line of the Simple Summary to “TMPRSS2”.